# High resolution observations of the ocean upper layer south of Cape São Vicente, western northern margin of the Gulf of Cadiz.

*Sarah A. Rautenbach[3], Carlos Mendes de Sousa[1,2], Mafalda Carapuço[4], Paulo Relvas[1]*

*1 Centre of Marine Sciences (CCMAR), University of Algarve, Faro, Portugal*

*2 Portuguese Institute for the Sea and Atmosphere (IPMA, I.P.), Lisboa, Portugal*

*3 Deltares, Delft, The Netherlands*

*4 Atlantic International Research Centre, Azores, Portugal*

**Abstract**

This article presents an Eulerian physical and biogeochemical data set from the Iberian Margin Cape São Vicente Ocean Observatory (IbMa-CSV), a facility of the European Multidisciplinary Seafloor and water column Observatory - European Research Infrastructure Consortium (EMSO-ERIC) located 10 nautical miles south of Cape São Vicente (Portugal), the southwest tip of the Iberian Peninsula and western limit of the northern margin of the Gulf of Cadiz. The observatory was installed on the shelf break, and the data time series spans four months for most of the variables. The upper 150 m were sampled intensively with a wave powered vertical profiler at an average rate of 4.5 profiles per hour recording at 2 Hz when ascending at approximate velocity of 0.2 m/s and 10 Hz when descending at variable velocity. The vertical resolution was always higher than 0.2 m. Measured channels were conductivity, temperature, pressure, chlorophyll *a*, dissolved $O_2$ concentration, and turbidity. Derived channels are sea pressure, depth, salinity, speed of sound, specific conductivity, dissolved $O_2$ saturation, density anomaly, spiciness and Brunt-Väisälä frequency. The acquired data set includes the flow velocity and direction along the water column, taken from an upward looking 300 kHz Acoustic Doppler Current Profiler (ADCP) recorded every hour for 3 m depth bins extending the same depth range of the vertical profiler. A standard quality control scheme was applied to the data set. The data set is preserved for multiple use and is accessible in the Sea Open Scientific Data Publication (SEANOE) repository, under the address: https://www.seanoe.org/data/00836/94769/ (Rautenbach et al., 2022).

**Key words:** High resolution dataset, vertical profiles, EMSO-ERIC, IbMa-CSV observatory, Cape São Vicente, Western Gulf of Cadiz

## 1. Introduction: the relevance of the site's location

The Iberian Peninsula (Figure 1) represents the northern branch of the Canary Current Upwelling System (CCUS), one of the four Eastern Boundary Upwelling Systems (EBUS), along with the Benguela, California and Humboldt or Peru upwelling systems. These systems are characterized by the coastal upwelling of cold nutrient rich subsurface water, driven by the joint action of northerly winds that blow at least during a substantial part of the year, and the Earth's rotation (Ekman mechanism). Therefore, those systems are among the most productive of the world ocean, which justifies their socio-economic relevance.

The CCUS is unique among the EBUS, since it is the only one punctuated by a discontinuity that is imposed by the entrance of the Mediterranean Sea into the Gulf of Cadiz (GoC) through the Strait of Gibraltar (Figure 1). The meridional coast of the western Iberian Peninsula is abruptly interrupted at the Cape São Vicente (CSV), the southwestern tip of the Iberian Peninsula. There, the coastline turns almost at a right angle into the zonal orientated northern margin of the GoC.

The continental shelf off the southern part of the western Iberia and in the CSV area is narrow (< 10 km wide south of 38º N), approximately delimited by the 200 m bathymetric contour. Over the continental shelf and slope, roughly from April to October, the oceanographic conditions are largely dominated by the upwelling process and associated cold jet flowing equatorward (Relvas and Barton, 2002). For the remainder of the year, the flow is expected to point northward, although there is a lack of observational evidence. Nevertheless, there is measured evidence that over the inner shelf the upwelling pattern is interrupted by the development of a warm coastal counter-current whenever the upwelling favorable winds relax below a certain threshold (Relvas and Barton 2005; Garel et al., 2016).

The Costal Transition Zone, defined as the region where the coastal waters interact with the offshore oceanic waters, is populated by a variety of mesoscale structures, such as meanders, eddies, and filaments. The CSV is the root of a recurrent upwelling filament that may extend more than 150 km offshore (Sanchez et al., 2008), exporting to the open ocean a much larger mass than expected by the purely wind-driven Ekman circulation. The new production of an upwelling season could be entirely exported to the open ocean by upwelling filaments (Arístegui et al., 2006), revealing the importance of such features to the ecosystem functioning.

At deeper levels, where the wind is not a forcing factor, the CSV region reveals fascinating processes related to the Mediterranean Outflow Water (MOW). After leaping the shallow sill (< 300 m deep) of the Strait of Gibraltar, the salty and warmer MOW sinks sharply into the deep GoC (depths up to 4000 m), and spreads at depths between 800 and 1200 m, where it finds the equilibrium in the gravitational field (Sanchez et al, 2017). However, a shallow vein detaches and flows at depths as shallow as 400 m or less along the northern continental slope of the GoC, turning poleward around the CSV (Ambar 1983; Cardeira et al., 2013).

The higher level of salt entering the North Atlantic through the Strait of Gibraltar and how it spreads throughout the Atlantic basin is a key factor with implications in the functioning of the Atlantic Meridional Overturning Circulation (AMOC), and therefore with climatic consequences. Due to the water column stability, diapycnal mixing of the MOW through entrainment occurs at long time scales when compared with horizontal dispersion through advection (Mauritzen et al, 2001). MOW is dominated by a succession of mesoscale rotating structures, the so called meddies (Mediterranean eddies) (Bower et al., 1995; Ambar et al., 2008). Meddies are described as rotating salt-water lenses, typically 50-200 km wide and 100-200 m thick. There is some evidence that the dynamic effect of meddies propagate along the entire water column, till the surface (Serra et al, 2010). CSV is identified as a site for meddy generation. Topographic features along the continental slope near CSV are hypothesized as meddy triggers. The key role that the CSV region plays in a wide variety of oceanographic processes of all scales, some impacting the entire North Atlantic circulation, demonstrates the relevance of the region to install a high-resolution subsurface observatory.

## 2. Motivation and Objectives

In the frame of the European Multidisciplinary Seafloor and water column Observatory – European Research Infrastructure Consortium (EMSO-ERIC – https://emso.eu/) physical and biogeochemical data from fixed ocean observation platforms throughout Europe are aggregated, harmonized, and shared openly under the Creative Commons Attribution License (CC-BY) license, guaranteeing open access for anyone. EMSO-ERIC is a distributed research infrastructure, encompassing observatories and test sites along European waters, from coastal to deep sea locations. Some observatories have already been operating for some time, whereas other nodes are yet to be established.

The EMSO-ERIC initiative defined the Iberian margin, specifically the region southwest of the CSV, as the location to install a regional facility of its European network. Along with other objectives related to geo-hazards seafloor observations, this was the opportunity to carry out long term *in situ* observations of the subsurface ocean in a clearly under sampled area, regarding its oceanographic relevance. In the region, *in situ* observations are limited to event scale records from research cruises. Therefore, the main goal was to construct continuous high resolution and long-term time series of oceanographic variables along the water column. A mobile platform carrying oceanographic sensors, moving continuously throughout the water column, robust enough to survive the energetic seas of the region for long periods, was carefully selected. The vertical definition of the flow field would be ensured by placing an acoustic doppler current profiler (ADCP) nearby, sampling the entire water column. The EMSO-Iberian Margin - Cape São Vicente observation platform (IbMa-CSV)is currently producing the first long term set of observations, from which the seminal deployment data are presented in this article.

While, *in situ* observations play a major role in understanding ocean dynamics and can be used for
various purposes, until today the availability of continuous and long-term *in situ* data of the ocean is
sparse. The construction of long high-resolution time-series is fundamental to access the long-term
physical and biogeochemical variability of the water column, and to improve modeling efforts, meeting
climatic change and ecosystem functions objectives. The data gathered are highly valuable for the
scientific community, with social and economic implications. Most political decisions are taken based
on evidence or future scenarios, mainly provided by numerical models. Due to the turbulent nature of
the ocean flow, numerical models need to be parametrized. More accurate parameterizations are
achieved when based on *in situ* observations, the higher the resolution the better, resulting in more
realistic numerical models. Therefore, one of the criteria that drove the choice of the observation devices
to install at IbMa-CSV was the generation of high-resolution records.
**3. Methods**
*3.1 The EMSO-Iberian Margin Cape São Vicente observatory (IbMa-CSV) - setup and operation*
The IbMa-CSV is located at the southwestern tip of the Iberian Margin, 10 nm south of CSV, on the
edge of the continental shelf (approximately 200 m depth). Deployment site selection carefully
considered fishing activity in the surrounding area, avoiding well known heavy equipment preferred
routes (e.g. trawling, longlines). A Permit for Private Use of the National Maritime Space (TUPEM)
was authorized by the Directorate-General for Natural Resources (DGRM) for an area of 0.35 km$^2$, in
which the observatory is deployed and should not be entered by other parties. However, ship traffic and
fishing activities pose a significant risk to the observatory as the TUPEM area is not patrolled. To
minimize this risk, engagement actions were undertaken with local communities and the legal
concession publicized in local navigation charts through official channels. This approach proved to be
successful as there was no visible and/or reported incident.   The boundaries of the TUPEM area are
N36°50.9087' W8°55.6243', N36°50.9600' W8°55.2200', N36°50.6800' W8°55.6800' and
N36°50.6800' W8°55.2200'. Within this area the instruments are fixed on three separate moorings
(Figure 1C). The TUPEM is managed by the Algarve Centre of Marine Sciences (CCMAR), Faro,
Portugal.

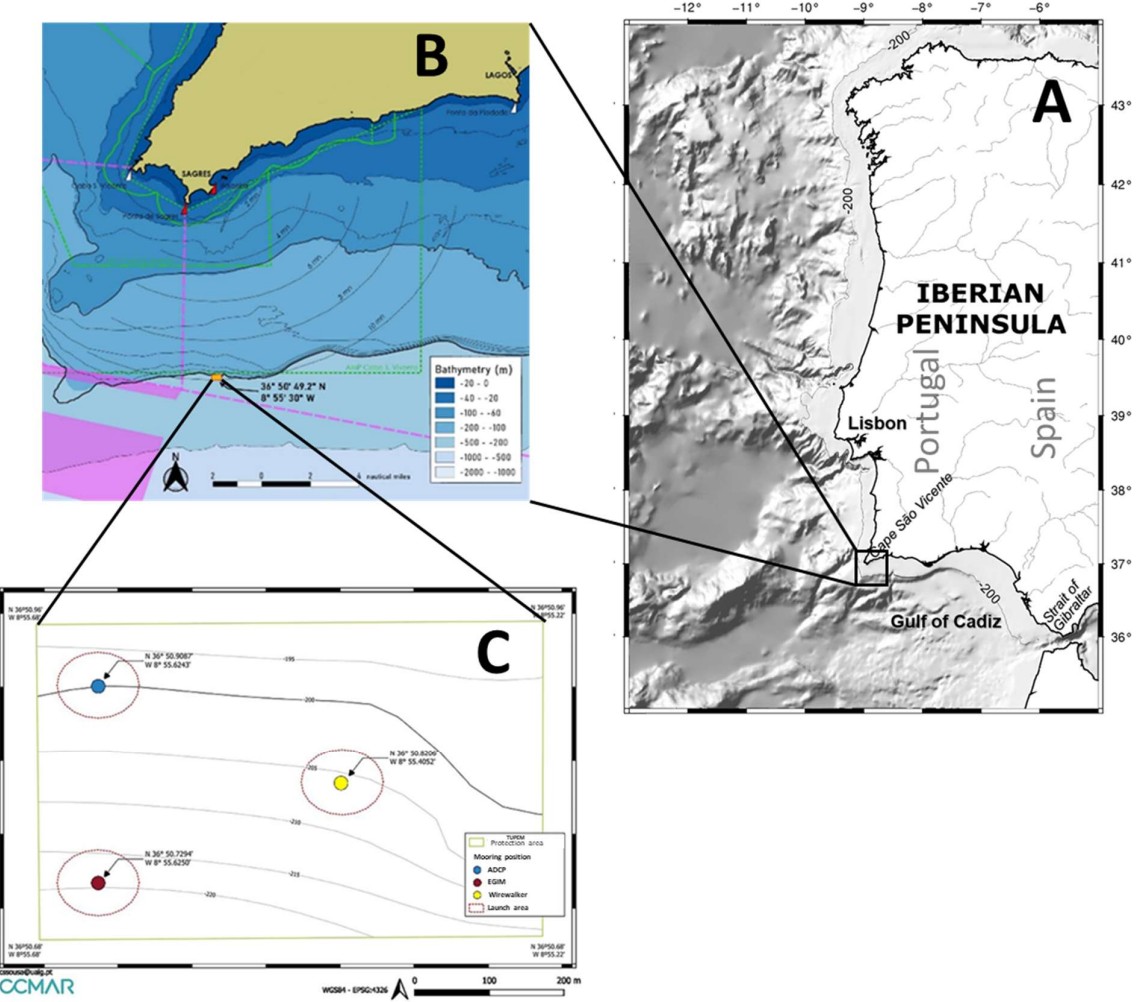


**Figure 1.** Location of the CSV region (**A**). Reserved TUPEM , managed by the CCMAR. IbMa-CSV is located within the TUPEM area (**B**). Mooring sites for each platform; vertical wave-powered profiler (surface, yellow), EGIM (subsurface, red), ADCP (subsurface, blue) (**C**).

Mooring design followed current best practices (e.g. Coppola et al., 2016), based on two platform types: subsurface (EGIM and ADCP), and surface (vertical profiler; Figure 2). The subsurface moorings were conceived as linear structures from anchor to buoy, while the surface mooring was based on an inverse catenary configuration. The choice of the hardware to be used in the mooring, i.e. the size and shape of the anchor, the type of rope and chain, number, size and shape of flotation aids and their position along the mooring line, linking hardware (shackles, swivels, d-links), were all carefully considered to meet the environmental features of the deployment area (e.g. waves, atmospheric forcing, presence of strong currents). Static and dynamic behavior of all three designs was then simulated in a dedicated software (Proteus DS), considering time dependent forcing parameters (wind, currents and waves) to evaluate vertical load, components position, tilt and tension, required safe anchor mass, and overall mooring configuration, according to different set scenarios, i.e. "normal", "storm", and "extreme".

Moorings were required not only to endure "extreme" conditions without failure, but also
maintain operational capabilities (to a reasonable extent) under more energetic events.
Simulated results pointed to neglectable instrumentation tilt of the subsurface moorings under
a set maximum 0.6 m/s current. Regarding the surface mooring, vertical travel wire inclinations
greater than 20º were expected to hinder vertical motion. Simulated inclinations were on
average 5.7º, 15.2º, and 29.5º, under "normal", "storm" and "extreme" scenarios, respectively,
as such considered to be satisfactory. Operating depths, i.e. the subsurface platforms placed
broadly below 150 m depth, and the vertical profiler travelling between approximately 150 m
and 1 m in constant motion (~5 or 6 full profiles per hour), were expected to deter significant
biofouling growth, requiring as such minor control techniques, such as a homespun coating
applied to the ADCP transducers (zinc oxide paste mixed with cayenne pepper), as well as
copper tape around the optical sensors. Recovered equipment experienced, as expected,
biofouling, however, while the ADCP subsurface platforms were unaffected, the vertical
profiler was compromised after the two months, where algae growth led to the salinity sensor
operation hinderance. Based on these findings, a new strategy must be developed for future
deployments, whether it is to clean the sensors regularly during the deployment period or using
innovative biofouling control techniques compatible with available sensors (wipers, non-toxic
coating, UV lights, for instance)
*3.2 Data acquisition platforms and settings Instruments, parameters, and sampling*
An ADCP (Teledyne RDI Sentinel V100 300 kHz), mounted in a upward facing subsurface buoy
(36.848478 N , -8.927072 E; 150 m), a vertical wave-powered profiler (Wirewalker) (36.84701 N, -
8.92342 E; near surface to 150 m), and an EMSO Generic Instrument Module (EGIM) (36.84549 N, -
8.927083 E; 200 m) were deployed from the R/V Mário Ruivo during the EMSO-PT Leg 2 Campaign,
in collaboration with the Portuguese Institute for the Sea and Atmosphere, I.P. (IPMA, I.P.), in the
TUPEM area during June – October 2022 (Figure 1C).
ADCP data were collected every hour for 3 m depth bins (51 bins in total), mounted at 150 m depth
(Figure 2). The blank right above the ADCP accounts for 2 m. Ping interval was 1 s and number of pings
120. East-west and north-south component (ms$^{-1}$) of the current together with the magnitude and
direction were acquired. Supplemental parameters, substantial for quality control, are provided
additionally, including correlation, echo intensity and percent of good return of each of the four beams,
as well as heading, pitch, and roll. The ADCP was further equipped with a thermistor and pressure
sensor. The ADCP was calibrated before the deployment according to manufacture guidelines.
A 6 channel RBRconcerto CTD, equipped with two Turner Designs Cyclops 7F sensors (Chl-a and
Turbidity) and one RBRcoda3 T.ODO (optical dissolved oxygen) were installed in a vertical wave-
powered vertical profiler , travelling from about 1 m below surface to 150 m depth at an variable speed:
upward cast (free floating) ~0.5 ms$^{-1}$; downward cast (wave motion) ~0.4 ms$^{-1}$ (depending on wave
conditions). Sampling rate was 2 Hz ascending, and 10 s descending. Measured channels were
conductivity, temperature, pressure, chlorophyll-a, dissolved $O_2$ concentration, and turbidity. Derived
channels are sea pressure, depth, salinity, speed of sound, specific conductivity, dissolved $O_2$ saturation,
and density anomaly.
The EGIM, equipped with a SeaBird SBE37, Aanderaa 4831dw, RBRquartz 3 BPR, WETLabs ECO-
NTU, OCEANSONICS icListen SB60L-ETH and Teledyne RDI 300kHz Workhorse Monitor direct-
reading ADCP, was fixed at approximately 200 m depth.
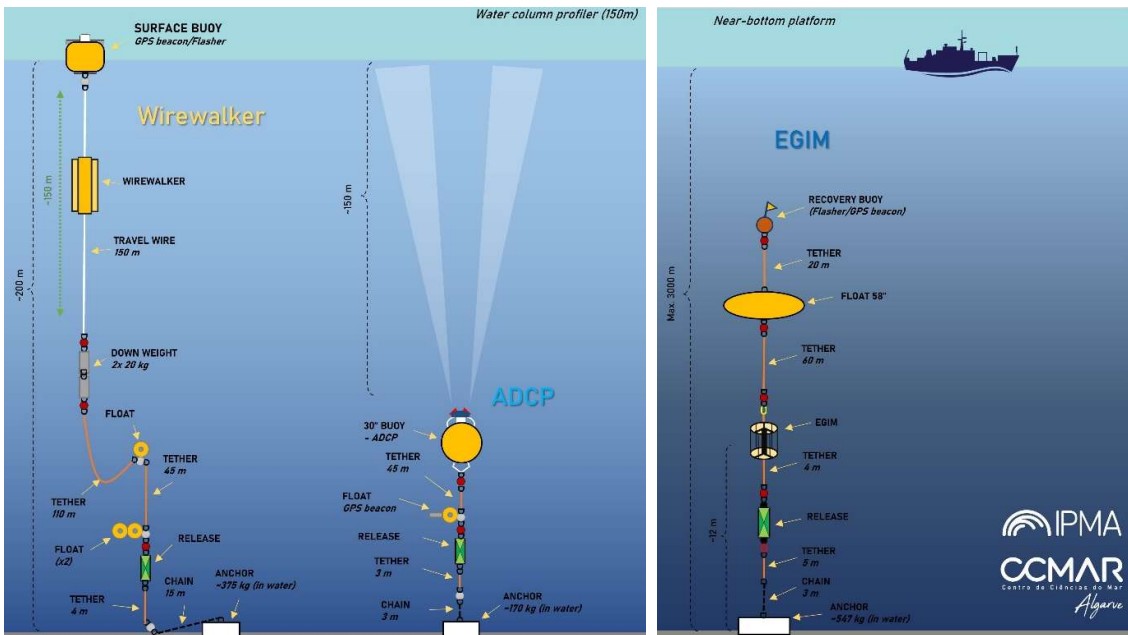

**Figure 2.** Schematic representation of the IbMa-CSV platforms. Left: Vertical wave-powered profiler and ADCP moorings,
managed by CCMAR. Right: EGIM mooring, managed by IPMA and CCMAR.
Sampling period was 60 minutes (ADCP), 15 minutes (CT, Turbidity, Oxygen), 30 seconds (Pressure),
and 5 minutes / 1 minute recording (acoustics). Measured channels include conductivity, temperature,
pressure, temperature, dissolved $O_2$ concentration, turbidity, currents, and passive acoustics. Derived
channels are sea pressure, depth, salinity, speed of sound, specific conductivity, dissolved $O_2$ saturation,
and density anomaly. The data time-series from the ADCP and the Wirewalker, managed by CCMAR,
will be presented in this data paper, along with the description of the data processing and results.
**4. Data files and metadata**
Instrument data files come in comma-separated value files and are converted into NetCDF format
according to CF Conventions 1.6. Files are named after facility code, platform code (WW, EGIM),
deployment number (D01, D02, …, Dnn), deployment period, and version (v001, v002, …, vnnn) e.g.
[folder_path]\IBMA-CSV_WW_D01_yyyymmdd_to_yyyymmdd_v001.nc. Changes are tracked in a
log-text file, which is located in the "dataset type" – directory. Instrument data (raw) are identified with
the code "*_v001" and metadata with code "*_M".
The vertical wave-powered profiler data are divided into six NetCDF files, each one approximately two
million data points, to keep file size reasonable. Each NetCDF is built upon the same structure: Global
attributes, Dimensions and Variables. Global attributes describe the dataset universally through a short
descriptive summary as well as other attributes such as temporal extension, geospatial position, principal
investigator, person of contact and more (Table 1). Each variable is embedded in one or more
dimensions, in this case: Time, Longitude, Latitude, Depth and Bins. Each parameter is accompanied
by a set of metadata attributes, holding detailed information about the instrument type, the units and
other relevant information regarding the variable. The SeaDataNet parameter discovery vocabulary
(https://vocab.seadatanet.org/search), well established in ocean science, is used for attributes,
dimensions, variables and units. Further, vocabulary is based on the Copernicus Marine Environment
Monitoring Service *In Situ* Thematic Assembly Centre (CMEMS INSTAC) Manual v3.2 and
SeaDataNet OceanSITES Data Format Reference Manual v1.4. Common vocabulary facilitates
machine-readability and manual findability by users. Each dataset is accompanied by comprehensive
metadata. Global and variable specific metadata attributes were agreed upon in the Data Management
Service Group (DMSG) of EMSO-ERIC (Table *1*). The main objective of EMSO-ERIC DMSG is to
make each dataset as findable, accessible, interoperable and (re)usable as possible, according to FAIR
standards, harmonize data quality control, format and metadata procedures. Each in this data paper
presented dataset can be reused under the CC-BY 4.0 license (https://spdx.org/licenses/CC-BY-4.0).
**Table 1.** EMSO-ERIC Data Management Service Group Metadata Catalogue.

| Global Attributes | Dimensions | Variables | Quality Control |
|---|---|---|---|
| date_created | long_name | long_name | long_name |
| Conventions | standard_name | standard_name | flag_values |
| institution_edmo_code | units | units | flag_meanings |
| institution_edmo_uri | axis | comment | conventions |
| insitution_ror_uri | ancillary_variables | coordinates | |
| geospatial_lat_min | sdn_parameter_name | ancillary_variables | |
| geospatial_lat_max | sdn_parameter_urn | sdn_parameter_name | |
| geospatial_lon_min | sdn_uom_name | sdn_parameter_urn | |
| geospatial_lon_max | sdn_uom_urn | sdn_parameter_uri | |
| geospatial_vertical_min | | reference_scale | |
| geospatial_vertical_max | | sdn_uom_name | |
| time_coverage_start | | sdn_uom_urn | |
| time_coverage_end | | sdn_uom_uri | |
| update_interval | | sensor_model | |
| site_code | | sensor_reference | |
| emso_facility | | sensor_type_uri | |
| source | | sensor_type_name | |

| | | | |
|---|---|---|---|
| platform_code | | sensor_manufacturer | |
| wmo_platform_code | | sensor_manufacturer_uri | |
| data_type | | sensor_serial_number | |
| format_version | | sensor_mount | |
| network | | sensor_orientation | |
| data_mode | | sensor_depth | |
| title | | QC_indicator | |
| summary | | | |
| keywords | | | |
| keywords_vocabulary | | | |
| project | | | |
| principal_investigator | | | |
| principal_investigator_email | | | |
| doi | | | |
| references | | | |
| license | | | |
| license_uri | | | |


Quality control variables were created for each measured parameter and for some derived parameters.
The quality control variable name is composed of the variable name of the parameter and the suffix
"_QC". Quality control procedures and flagging conventions are described in further detail in the next
section. For each dataset it was assured that solely measurements conducted in the water column were
considered. This was achieved by examining depth measurements, derived from the pressure sensor as
well as temperatures indicating atmospheric temperatures. Out of water values were removed from each
dataset.
**5. Technical Validation**
Each dataset was subject to quality control (qc). Suspicious and bad values were not removed from the
published raw dataset. Instead, a complimentary qc-variable was created, holding flag values describing
each individual parameter value. Flag values are defined by the OceanSITES Data Format Reference
Manual v1.4 (OceanSITES, 2020). Flag may take the values 0, 1, 2, 3, 4, 7, 8, 9 that are defined as
"unknown", "good_data", "probably_good_data", "potentially_correctable bad_data", "bad_data",
"nominal_value", "interpolated_value", "missing_value", respectively. Suspicious and bad values were
flagged as "potentially_correctable bad_data" (3) and "bad_data" (4), respectively.
Contrary to the published raw dataset, this paper presents the quality controlled data. Data flagged as
"potentially_correctable bad_data", "bad_data" and "missing_value" were excluded from the plots
presented in this manuscript.
ADCP quality control was based on the quality control procedures from Garel *et. al*. (2016). To ensure
that no data subject to site lobe interference is shared, the upper 10 % of the data was flagged as
"bad_data". Further, the sea surface was detected by locating the cells with a difference among adjacent
values greater than three and flagged bad accordingly. This criterion was restricted to cells above the
14[th] cell (100 m) to prevent misinterpretation of the surface. Cells above and the cell immediately below
were flagged as "bad_data". Furthermore, if two or more beams with cells featuring a difference among
adjacent bins in echo intensity > 30, and/or with three or more out of four beams with correlation
magnitude values lower 64 counts were also flagged  as "bad_data". Temperature was controlled
according to SeaDataNet Guidelines (see above), and pressure was assessed via visual inspection.
The first quality control check for the vertical profiler data was done visually via line and boxplots of
each variable, allowing a global and regional range check and spike detection at first sight. Quality tests
applied on each variable of this dataset were: Sensor range test, global range test, regional range test and
spike test. A gradient test was additionally applied to temperature and salinity. Global ranges were
obtained from literature for each variable, whereas regional ranges were discussed and selected with the
support of experts from the region.
Temperature and salinity Spike Test (ST) was conducted according to SeaDataNet Data Quality Control
Procedures Manual (SeaDataNet 2010), using the following algorithm: Test value = | V2 - (V3 + V1)/2
| - | (V3 - V1) / 2 | > V THRESHOLD. The value is flagged "bad_data" when the test value exceeds
6 °C, 0.9 PSU, respectively. Gradient Test (GT) relied on the following from SeaDataNet proposed
algorithm: Test value = ( | V2 - (V1 + V3)/2 | > V_GRAD. The value is flagged "bad_data" when the
test value exceeds 9 °C, 1.5 PSU, respectively. Spikes in conductivity were determined by Interquartile
Range Test (IQR) (Hald, 1952). Quartile two and quartile three make up the interquartile range (IQR)
of the data. Two thresholds are defined for "suspicious" (1.5) and "bad_data" (3). The IQR is multiplied
by each threshold and subtracted (added) from quartile 1 (quartile 3). If a data point exceeds the
computed range, it is flagged accordingly. A IQR test was not applied on other variables as it was found
to be overly sensitive to biogeochemical variables, discarding reasonable values. Therefore, other
manuals and standards were used for spike detection in biogeochemical parameters.
Dissolved oxygen, alongside with oxygen saturation, were assessed based on the ST proposed in the
Manual for Real-Time Quality Control of Dissolved Oxygen Observations by the Integrated Ocean
Observing System (IOOS) Quality Assurance / Quality Control of Real Time Oceanographic Data
(QUARTOD) (IOOS QUARTOD, 2018). A spike reference (average of adjacent points $DO_{n-2}$ and $DO_n$)
is subtracted from the tested value ($D_{n-1}$) and tested against an upper and lower threshold. Values failing
the upper boundaries are flagged as "bad_data", values in the range of the lower and upper threshold are
flagged as "suspicious". Thresholds for dissolved oxygen and oxygen saturation were set at 4 mgl-1
(lower) and 8 mgl-1(upper) and 80 % (lower) and 120 % (upper), respectively. The most reliable
chlorophyll-a spike detection for this dataset is proposed by The Platforms for Biogeochemical studies:
Instrumentation and Measure (PABIM) (PABIM, 2010). The ST algorithm remains the same as in the
SeaDataNet Guidelines for temperature and salinity. PABIM (PABIM, 2010) suggests an algorithm to
define the threshold value, most appropriate in any region, which is computed as follows:
Threshold_Value = |median(V0,V1,V2,V3,V4)| + |σ (V0,V1,V2,V3,V4)|. Turbidity spikes are detected
with the same methodology as chlorophyll-a, using a predefined threshold of 6 NTU.

## 282     6. Data Records

In this section we visualize the entire data series of the vertical profiles of the of the measured and
derived variables in a comprehensive way. Only validated data are displayed. Data considered as bad or
potentially bad were not considered to display or for the interpolations, as stated in the previous section.
Preliminary analyses as well as basic statistics are presented.
*6.1 Acoustic Doppler Current Profiler*
Current data from the upward facing ADCP were acquired from June to October 2022 at a depth of 150
m (Figure 3). Measurements above 10 m failed the quality control criteria due to interference with the
surface resulting in biased data and were discarded. In the plots we present only the data below 20 m
deep (Figure 3).

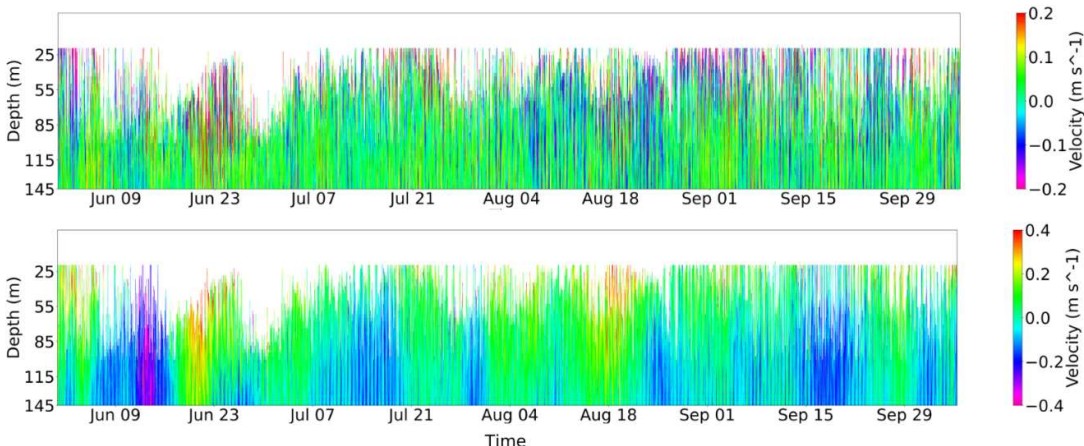


**Figure 3.** Meridional (north-south) (**top**) and zonal (east-west) (**bottom**) component of acoustic doppler current profiler
throughout the whole water column from June to October 2022. White patches reveal absence of valid data. Negative values
indicate southward (westward) flow, whereas positive values indicate northward (eastward) flow direction. Measurements are
expressed in ms$^{-1}$. Data above approximately 20 m suffer from surface interference and were removed.

Current meter records demonstrate an energetic current regime in the area south of CSV. Clearly, the
dominant flow is zonal. The meridional component is weak, without any clear tendency in the direction
(notice the different scales of the velocity in Figure 3). The zonal flow shows a prevailing eastward flow,
interleaved with sudden inversions to westward. Westward flows are more frequent towards the seafloor.
Current meter records were divided into three depth ranges to understand the distinctive current regimes
along the vertical. The upper layer (UL) comprises the surface waters, reaching down to 60 m. The
middle layer (ML) of the water column ranges from 60 to 100 m, and the bottom layer (BL) covers from
100 to 150 m. Polar plots were created for each depth layer to depict the vertical change of the magnitude
and direction of the flow (Figure 4). A relatively energetic flow, showing a few episodes of increased
velocities > 0.75 m s$^{-1}$, prevails in the upper layer. There, the flow shows a strong eastward component,
contrasting with the almost absence of westward flow. In the middle layer this prevalence diminishes,
and the flow intensity decays, with velocities sporadically reaching values > 0.6 m s$^{-1}$, but mostly ranging
between 0.001 – 0.4 m s$^{-1}$. As we approach the seafloor, in the bottom layer, the flow is weak, with
velocities between 0 and 0.2 m s$^{-1}$, and a prevailing westward component is evident, opposed to the
upper ocean layer. A basic statistic of the flow velocity for each depth interval is presented too.

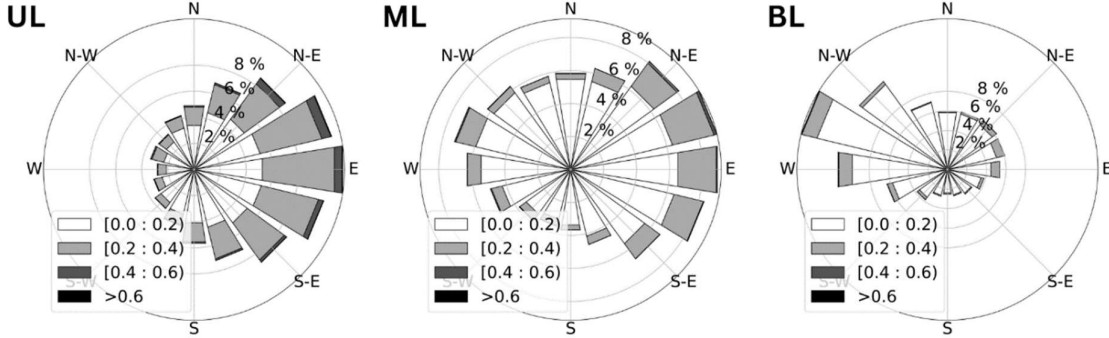


**Figure 4.** Demonstration of current magnitude and direction of the upper layer (UL; 10 - 60 m), middle layer (ML; 60 – 100m), bottom layer (BL; 100 – 150 m).

To detail the temporal variability of the mean flow in each depth layer, stick diagrams are presented for
each depth layer (Figure 5). The intensified current in the upper layer can be observed throughout the
whole period of the deployment. A more diffuse pattern in direction, along with the decrease in velocity
can be observed in the lower layers, except for a short period during mid-June. However, there is a
prevalence of zonal flow, interrupted periodically by momentary direction changes. June can be
identified as the most energetic month in the time series, featuring the highest mean values throughout
the water column.

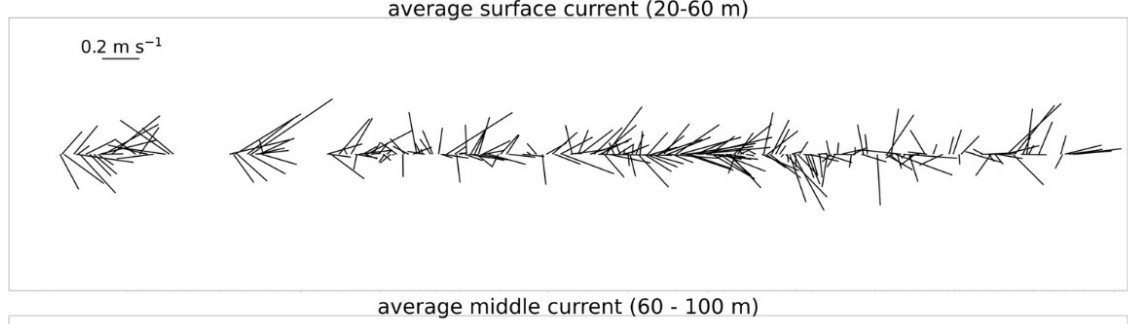

average surface current (20-60 m)

0.2 m s⁻¹

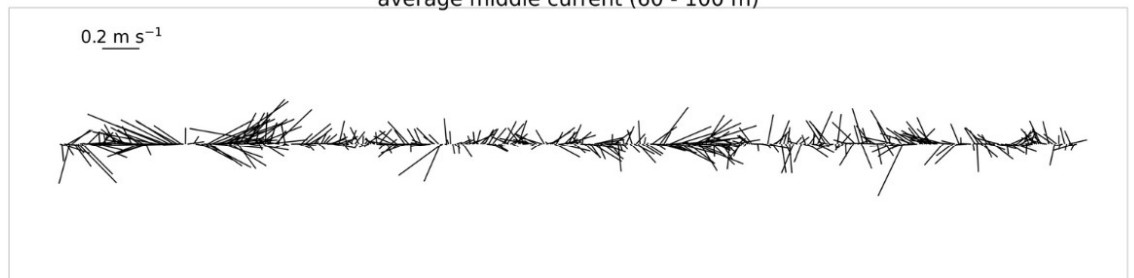

average middle current (60 - 100 m)

0.2 m s⁻¹

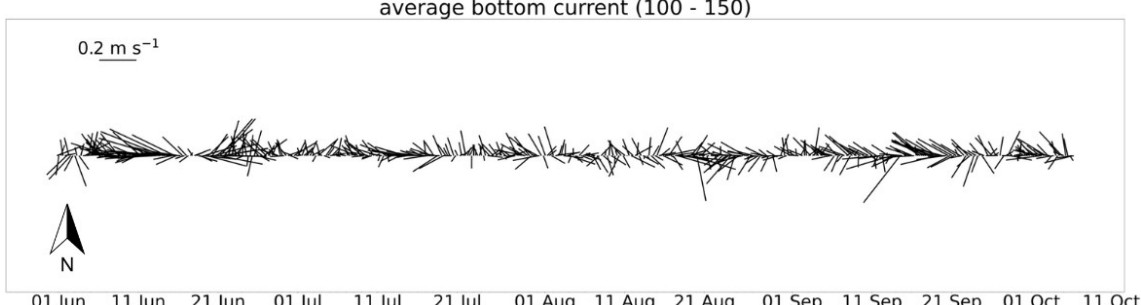

average bottom current (100 - 150)

0.2 m s⁻¹

N

01 Jun  11 Jun  21 Jun  01 Jul  11 Jul  21 Jul  01 Aug  11 Aug  21 Aug  01 Sep  11 Sep  21 Sep  01 Oct  11 Oct
**Figure 5.** ADCP stick plot from June to October 2022 divided into upper layer (20-60 m; top), middle layer (60-100 m; middle)
and bottom layer (100-150 m; bottom).
**Table 2.** Statistics of upper layer (UL; 10-60 m), middle layer (ML; 60-100 m) and bottom layer (BL; 100-150 m) grouped
by month. SD representing the standard deviation. Values are expressed in ms⁻¹.

| | UL | | | | ML | | | | LL | | | |
|---|---|---|---|---|---|---|---|---|---|---|---|---|
| | Mean | SD | Min | Max | Mean | SD | Min | Max | Mean | SD | Min | Max |
| **June** | 0.247 | 0.105 | 0.009 | 0.773 | 0.186 | 0.095 | 0.004 | 0.574 | 0.142 | 0.080 | 0.001 | 0.687 |
| **July** | 0.142 | 0.078 | 0.002 | 0.730 | 0.100 | 0.059 | 0.001 | 0.787 | 0.089 | 0.047 | 0.000 | 0.509 |
| **August** | 0.182 | 0.096 | 0.001 | 0.639 | 0.125 | 0.074 | 0.000 | 0.469 | 0.098 | 0.053 | 0.001 | 0.385 |
| **Sept** | 0.151 | 0.096 | 0.003 | 0.716 | 0.107 | 0.059 | 0.001 | 0.600 | 0.101 | 0.058 | 0.002 | 0.510 |

*6.2 Vertical wave-powered profiler*
Continuous time series of the entire water column are highly valuable as they offer vast amounts of data
and can create a comprehensive picture of mesoscale and sub-mesoscale processes. The vertical wave-
powered profiler, equipped with physical and biogeochemical sensors, operated for four months
continuously, and delivered a rich dataset at the end of the deployment. Vertical profiles of the water
column show temperatures between 12.5 °C closer to the seafloor to approximately 22 °C on the surface
(Figure 6; Table *3*). The thermocline remains between 20-40 m, showing some periods of a well-mixed
homogenous surface layer and periods of more stratified waters (Figure 6; Figure 7). Salinities are found
to range between 33 and 37 from June to the end of July with an average salinity of 35.95 (SD ±0.13)
and 35.88 (SD ±0.09), respectively (Figure 6; Table 3). Salinity data beyond that point were discarded
and will not be discussed further as the conductivity sensor was subject to intense biofouling, prohibiting
the collection of trustworthy measurements after the month of July. Due to that the mixed layer depth
was only computed for the months of June and July, showing an average Mixed-Layer Depth (MLD)
around the 10-20 m mark, following the pattern of the thermocline (Figure 8, top). The dissolved oxygen
sensor shows lower oxygenated waters in deeper waters but stopped operating after two weeks
(Figure 6). The chlorophyll-a maximum can be found between 20-60 m with concentrations between
1-10 mg m$^{-3}$ and mitigates to almost 0 mg m$^{-3}$ below (Figure 6; Figure 7; Table 3). Turbidity
concentrations correspond to chlorophyll-a during the whole course of the measurements, with average
concentrations of 0.25 NTU (SD ±0.17), indicating the correlation between turbidity and biomass with
some additional phases of increased turbidity concentrations close to the seafloor (Table *3*).

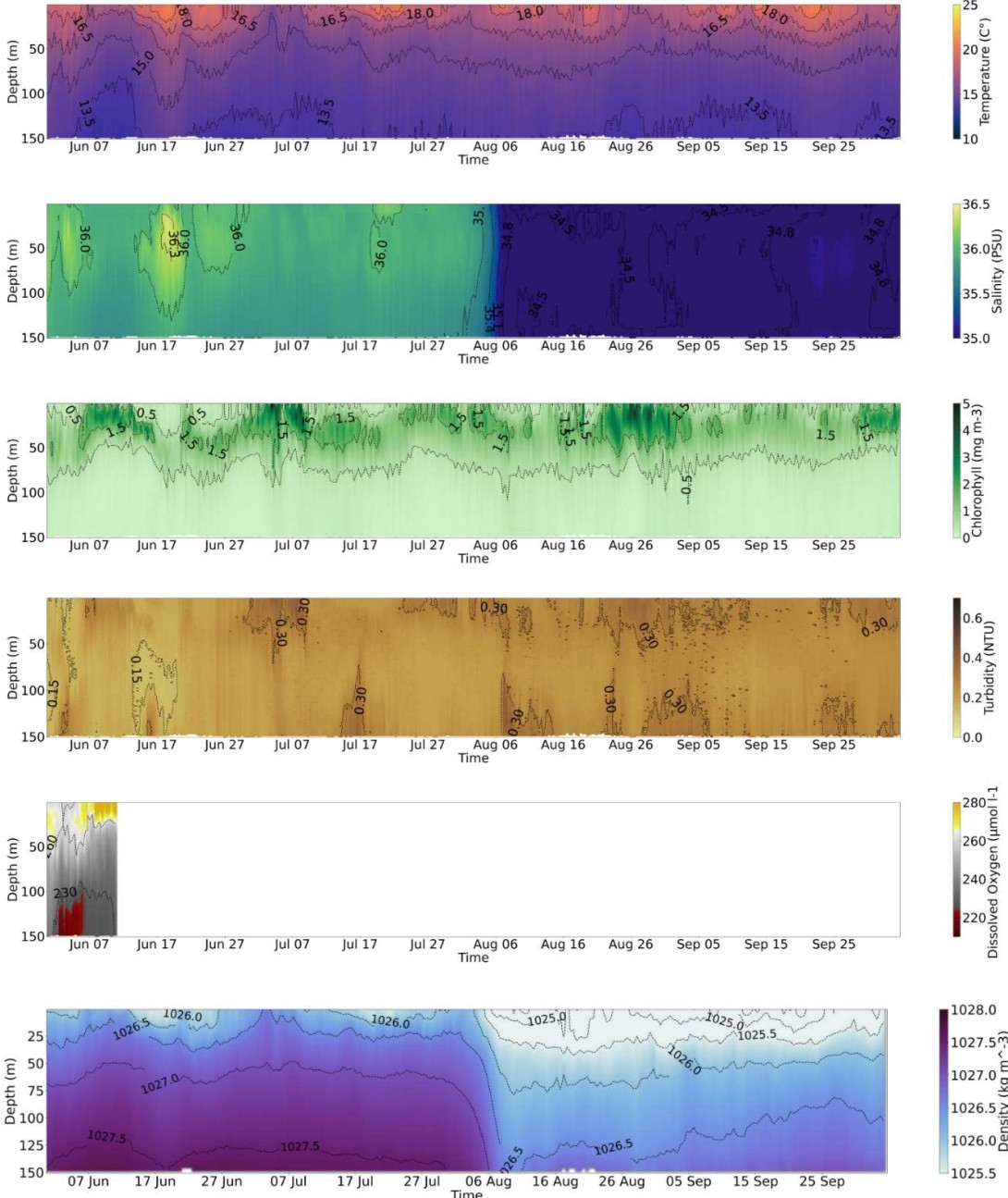

**Figure 6.** Continuous vertical wave profiler data Jun-Oct 2022. **Top to bottom**: Temperature (°C), salinity, chlorophyll-a concentration (mg m$^{-3}$), turbidity (NTU), dissolved oxygen (µmol l$^{-1}$), and density (kg m$^{-3}$). Salinity and density values after the month of July are ambiguous and are discarded from further discussion.

In June two period of increased salinity were recorded between the 1$^{st}$ – 6$^{th}$ and 13$^{th}$ to the 21$^{st}$ of June near the surface down to 120 m depth, together with mitigation in chlorophyll-a concentration, migrating to deeper layers along the mixed layer to a depth of approximately 60 m (Figure 6; Figure 8). The salty waters appear in form of an isolated lens, carrying maximum salinities of 36.72 (Figure 6; Figure 7). Simultaneously, an intensification in stability and spiciness can be observed (Figure 8). Spiciness was

computed with the Thermodynamic Equation of Seawater 2010 (TEOS-10) from Absolute Salinity and
Conservative Temperature, according to McDougall and Krzysik (2015).
Table 3. Statistics of vertical wave-powered profiler parameters grouped by month. SD representing the standard deviation.
Recordings are lacking for the variables dissolved oxygen, oxygen saturation after the first half of June and for salinity after
July.

| | June | | | | July | | | |
|---|---|---|---|---|---|---|---|---|
| | Mean | SD | Min | Max | Mean | SD | Min | Max |
| Temperature (°C) | 15.55 | 1.97 | 12.62 | 21.3 | 15.15 | 1.7 | 12.64 | 20.51 |
| Conductivity (S m⁻¹) | 4.46 | 0.21 | 4.13 | 5.06 | 4.41 | 0.18 | 4.14 | 4.98 |
| Salinity | 35.95 | 0.13 | 34.44 | 36.72 | 35.88 | 0.09 | 34.13 | 37.02 |
| Dissolved Oxygen (µmol l⁻¹) | 245.35 | 18.5 | 209.96 | 305.46 | - | - | - | - |
| Oxygen Saturation (%) | 97.77 | 10.5 | 81.08 | 120.0 | - | - | - | - |
| Chlorophyll a (mg m⁻³) | 0.59 | 0.65 | 0.08 | 9.33 | 0.76 | 0.81 | 0.09 | 10.84 |
| Turbidity (NTU) | 0.2 | 0.07 | 0.0 | 6.17 | 0.25 | 0.09 | 0.0 | 6.42 |
| Sound velocity (m s⁻¹) | 1510.48 | 5.43 | 1502.08 | 1526.23 | 1509.22 | 4.62 | 1502.31 | 1524.2 |
| | August | | | | September | | | |
| | Mean | SD | Min | Max | Mean | SD | Min | Max |
| Temperature (°C) | 15.41 | 1.64 | 12.64 | 21.56 | 15.48 | 2.03 | 12.74 | 21.22 |
| Conductivity (S m⁻¹) | 4.31 | 0.16 | 4.01 | 4.88 | 4.32 | 0.19 | 4.02 | 4.90 |
| Salinity | - | - | - | - | - | - | - | - |
| Dissolved Oxygen (µmol l⁻¹) | - | - | - | - | - | - | - | - |
| Oxygen Saturation (%) | - | - | - | - | - | - | - | - |
| Chlorophyll a (mg m⁻³) | 0.83 | 0.88 | 0.09 | 10.51 | 0.61 | 0.58 | 0.08 | 9.40 |
| Turbidity (NTU) | 0.27 | 0.10 | 0.04 | 7.13 | 0.27 | 1.41 | 0.03 | 363.77 |
| Sound velocity (m s⁻¹) | 1508.76 | 4.35 | 1500.69 | 1524.31 | 1508.89 | 5.44 | 1501.0 | 1524.69 |


Throughout the first half July colder and less saline waters shoal towards the surface with average values
of 14.83 ± 1.44 °C and 35.87 ± 0.07, respectively (Figure 6), resulting in a well-mixed and homogenous
water column. In response, stability and spiciness decrease (Figure 8), accompanied by increasing
chlorophyll-a concentration (0.9 ± 1.03 mg m⁻³) and turbidity (0.26 ± 0.07 NTU). In the second half of
the months surface waters experience warming and average increase slightly up to 15.40 ± 1.83 °C,
along with an increased stabillity and spiciness (Figure 6; Figure 8). Stratification enhances during
August due to a deepening of the warmer surface waters to a depth of approximately 60 m (Figure 8).
Around the 24th of August colder temperate waters shoal towards the surface (19.4 °C), simultaneously
with an inflation of the maximum chlorophyll-a concentration (10.51 mg m⁻³), attenuating in the
beginning of September. Upper layer stratification stabilizes throughout the month of September, with
temperatures around 21 °C in the upper 40 m with an increased period of warming between the second
and third week of the month along with a slight decrease of chlorophyll-a (Figure 6). The same pattern
was observed during mid-August, in which, with increased surface temperatures, higher chlorophyll-a
concentrations migrate to deeper layers, similar to the third week of July.

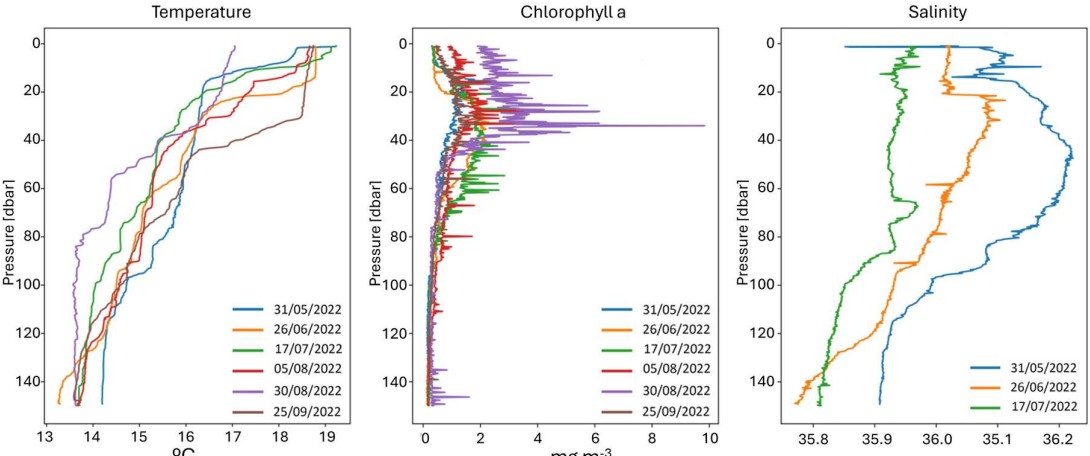

**Figure 7.** Examples of individual vertical profiles of temperature (C°) (**left**), chlorophyll-a (mgm-3) (**middle**) and salinity (**right**).

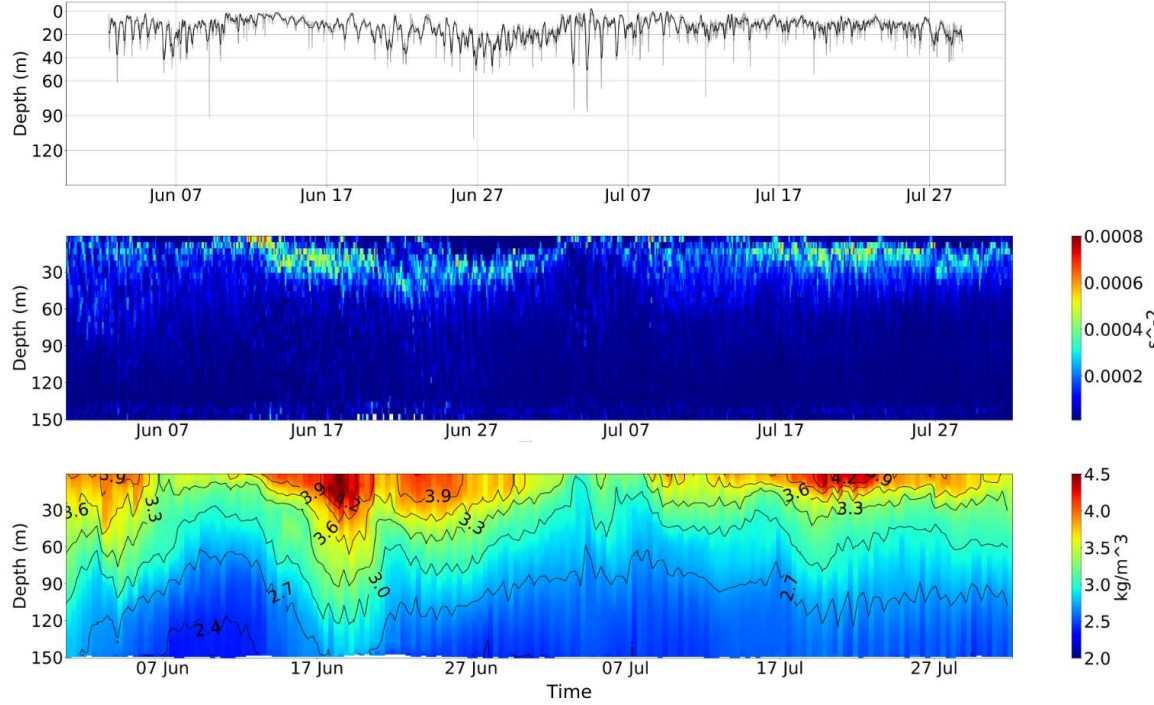

**Figure 8.** From **top to bottom**: MLD, Brunt-Väisälä frequency (N2), spiciness. Computed for the months of June and July. Subsequent failure of conductivity sensor prohibits computation of presented parameters from that point on.

## 7. Data availability

Quality controlled datasets are made publicly available as NetCDF files at the environmental data repository SEANOE (https://www.seanoe.org/) under the DOI https://doi.org/10.17882/94769 in accordance with FAIR principles (Wilkinson et al., 2019). Beyond the repository, data is ingested into the CCMAR Erddap server (https://erddap.ccmar.ualg.pt/erddap/index.html), in which a first data visualization and data can be downloaded in various file formats selectively from the user. Further, the data is shared with the EMSO-ERIC Data Portal (https://data.emso.eu/home), in which users can

visualize and download data according to their needs. The data is not restricted and is accessible for
anyone, accompanied with comprehensive metadata. Standardized datasets allow machine readability
and interoperability with various software.
**8. Data set value**
This dataset conveys the importance of continuous, long-term data acquisition and ocean monitoring to
capture mesoscale and sub-mesoscale events in the ocean. As presented before it was detected, for
example during the second half of June, fascinating thermohaline records. This deployment was the first
test run of the IbMa-CSV Ocean Observatory. Sensor failure due to biofouling will be tackled by
following a regular cleaning of the sensors at specific time intervals while deployed (profiler), and by
reducing deployment turn-around with a second vertical wave-powered profiler. Hence, the two profilers
will alternate in a minimum four month rhythm, therefore guarantying a continuous data collection. The
vertical wave-powered profiler offers impeccable high temporal and vertical resolution data products at
reasonable cost and maintenance. The only instruments which provide data products with comparable
resolution are autonomous underwater vehicles and gliders. Yet both economically and regarding the
scope of establishing an Eulerian, long-term observation platform, these instruments cannot compete,
underpinning the exceptional potential of the vertical wave-powered profiler and its data products.
The monitoring of energetic areas, such as the western tip of the northern margin of the Gulf of Cadiz
(the CSV), is crucial to understand the complexity of the ocean dynamics and to predict future
development via ocean models and their validation through comprehensive datasets. A wide range of
processes, from the upper layers wind induced upwelling to deeper MOW features, do occur in the ocean
surrounding CSV, as described in the Introduction. Intense mesoscale and sub-mesoscale activity, that
represent the "weather" variability of the ocean imposed by the turbulent nature of the circulation, are
quite conspicuous in this region and dominates all levels of the water column, challenging the
investigation of a wide range of oceanographic processes.
Efforts have been made to develop numerical models for this region, with the aim of better
understanding the exchange and mixing processes that occur there, and their implications for the
ecosystem and salt spreading in the North Atlantic. However, there is no general theory of turbulence,
and numerical models must rely on parametrizations to solve this macro-turbulence. The correct
parameterization of the turbulent behavior of the ocean depends on the previous knowledge that we have
about the physical characteristics of the region to be modelled. This knowledge is built upon the
observation of the ocean. Higher resolution observations will produce better parameterizations of the
numerical models. The present knowledge of the oceanography of the region is inferred from event scale
sampling, leading to  regional numerical models highly data deficient., that tend to use parameterization
analogies with ocean regions with similar oceanographic characteristics and intensively sampled, such
as the California Upwelling System (Macias et al., 2014; Janeiro et al., 2017). This data series will make
it possible to find better parameters for the region and to solve more realistically the turbulence and
turbulence related ocean processes.
The present data set, with such vertical and temporal resolution, is unique in the region. The nearest
moorings, operated by the Instituto Hidrográfico (Portugal), are more than 120 km away, and only take
measurements at the surface (https://www.hidrografico.pt/index/en). The velocity field is assessed only
at the surface through HFRs that cover the region, operated by the Puertos del Estado (Spain)
(https://www.puertos.es) and the Instituto Hidrográfico (Portugal). For the first time the subsurface is
sampled in the region. The high-resolution sampling, covering the surface layer down to a depth of 150
meters, makes this dataset unique in a vast area of the ocean, disclosing the high oceanographic value
of the data set. The IbMa-CSV Ocean Observatory was established in the scope of EMSO-ERIC, a
European wide ocean observatory network, and will be further developed and improved to operate
continuously and long-term.

## 9. Usage note

EMSO data are published without any warranty, express or implied. The user assumes all risk arising
from the use of EMSO data. EMSO data are intended to be research-quality and include estimates of
data quality and accuracy, but it is possible that these estimates or the data themselves contain errors. It
is the sole responsibility of the user to assess if the data are appropriate for his/her use, and to interpret
the data, data quality, and data accuracy accordingly. EMSO welcomes users to ask questions and report
problems to the contact addresses listed in the data files or on the EMSO web page.

## 10. Acknowledgement

This study received Portuguese national funds from: FCT - Foundation for Science and Technology
through project UIDB/04326/2020, UIDP/04326/2020 and LA/P/0101/2020; operational programmes
CRESC Algarve 2020 and COMPETE 2020 through projects EMBRC.PT ALG-01-0145-FEDER-
022121 and EMSO-PT ALG-01-0145-FEDER-022231; EEA Grants Blue Growth project "Atlantic
Observatory – Data and Monitoring Infrastructure" (PT-INNOVATION-0002). Furthermore, we would
like to acknowledge and thank the R/V Mário Ruivo for the ship-time and support of the crew. The
authors would like to acknowledge that the research for this study was conducted at the Centre of Marine
Sciences (CCMAR), University of Algarve, Faro, Portugal and the Portuguese Institute for the Sea and
Atmosphere (IPMA, I.P.), Lisboa, Portugal.

## 11. Author Contribution

CS designed the instrument setup and the deployment strategy. SR, PR, CS, MC carried out the
equipment deployment and recovery. SA did the data processing, harmonization and publishing. SR,
PR, CS prepared the paper, with contributions from all co-authors.

## 12. Competing Interests

The contact author has declared that none of the authors has any competing interests.

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
