# Peer review of "https://doi.org/10.5194/es"

_Earth System Science Data, 2023_

## Author Response (AR1)

**Author's Response**

REVIEWER 1

Review of "High resolution observations of the upper ocean layer .....)

Summary:  As a report on the efforts to make observations as well as a preliminary look at the observations, some content was lacking and suggestions are made here for a revision.

The authors are deeply grateful to the reviewer for the effort put into this review. We truly appreciate the detailed and helpful review. It will contribute to improving the manuscript.

As a first comment to the overall review, that will be detailed below, the presented set of observations corresponds to an observatory in the frame of the EMSO-ERIC consortium (https://emso.eu/), that the authors want to keep as "permanent" as possible. The observatory is "science driven", but not "feature driven". The presented time series is not part of an "experiment" focused on any mesoscale structure. The goal of the time-series construction is long-term, not event scale.

Reviewer: Goals.A section 2 laying out the goals for this observatory would be a great addition.  The introduction lays out the motivation and the methods section shows the hardware, but what were the goals or objectives that drove the choice of the observing tools.  Then, later in the paper, one could discuss whether or not the goals were met and how the effort might evolve to have greater success at reaching the goals.  As a concrete example, is observing the velocity across the full water column a goal?  If so, might in the future a current meter be added to the wave driven profiler? Or, observing the velocity over the full depth is not a goal, so the failure of the ADCP to capture velocity near the surface is not a failing?  What is the intended longevity of the observatory?  forever?  Is this paper discussing just one deployment of an intended long term deployment?

We accept the reviewer's suggestion and decide to clarify the goals of the observatory in a new section 2 – Motivation and Objectives.

The last paragraph of section 1. Introduction: the relevance of the site location. already partially described the observatory's objectives. We decided to transfer this paragraph, slightly modified, from section 1. to the new section 2., and elaborate about the objectives, and frame them. The new section is:

2 – Motivation and Objectives.

In the frame of the European Multidisciplinary Seafloor and water column Observatory – European Research Infrastructure Consortium (EMSO-ERIC – https://emso.eu/) physical and biogeochemical data from fixed ocean observation platforms throughout Europe are aggregated, harmonized, and shared openly under the CC-BY license, guaranteeing open access for anyone.

EMSO-ERIC is a distributed research infrastructure, encompassing observatories and test sites along European waters, from coastal to deep sea locations. Some observatories have already been operating for some time, whereas other nodes are yet to be established.

The EMSO-ERIC initiative defined the Iberian margin, specifically the region southwest of the CSV, as the location to install a regional facility of its European network. Along with other objectives related to geo-hazards seafloor observations, this was the opportunity to carry out long term *in situ* observations of the subsurface ocean in a clearly under sampled area, regarding its oceanographic relevance. In the region, *in situ* observations are limited to event scale records from research cruises. Therefore, the main goal was to construct continuous high resolution and long-term time series of oceanographic variables along the water column. A mobile platform carrying oceanographic sensors, moving continuously throughout the water column, robust enough to survive the energetic seas of the region for long periods, was carefully selected. The vertical definition of the flow field would be ensured by placing an acoustic doppler current profiler (ADCP) nearby, sampled the entire water column. The EMSO-**Ib**erian **Ma**rgin - **C**ape **S**ão **V**icente observation platform (IbMa-CSV) is currently producing the first long term set of observations that are presented in this article.

Yet, *in situ* observations play a major role in understanding ocean dynamics and can be used for various purposes, until today the availability of continuous and long-term *in-situ* data of the ocean is sparse. The construction of long high-resolution time-series is fundamental to access the long-term physical and biogeochemical variability of the water column, and to improve modeling efforts, meeting climatic change and ecosystem functions objectives. The data gathered are highly valuable for the scientific community, with social and economic implications. Most political decisions are taken based on evidence or future scenarios, mainly provided by numerical models. Due to the turbulent nature of the ocean flow, numerical models need to be parametrized. More accurate parameterizations are achieved when based on *in situ* observations, the higher the resolution the better, resulting in more realistic numerical models. Therefore, one of the criteria that drove the choice of the observation devices to install at IbMa-CSV was the generation of high-resolution records.

Reviewer: Methods - clear statement of goals would help the reader understand why the instruments were fitted with the sensors listed.

We believe this comment was answered with the introduction of Section 2. Fitted sensors were those considered essential to a physical and biochemical characterization of the water column, along with the manufacturer's availability.

Reviewer: Methods - not found but expected here is a discussion of how the mooring designs were chosen. Just to survive expected flows but OK to lean over and get pushed down by drag?  Or, designed to be as close as vertical at all times by adjusting float buoyancy?  How much do the subsurface moorings get depressed by current?

Mooring design followed current best practices (e.g. Coppola et al., 2016)[1], based on two platform types: subsurface (EGIM and ADCP), and surface (vertical profiler). The subsurface moorings were conceived as linear structures from anchor to buoy, while the surface mooring was based on an inverse catenary configuration.

The choice of the hardware to be used in the mooring, i.e. the size and shape of the anchor, the type of rope and chain, number, size and shape of flotation aids and their position along the mooring line, linking hardware (shackles, swivels, d-links), were all carefully considered to meet the site environmental features (e.g. waves, atmospheric forcing, presence of strong currents) of the deployment area.

Static and dynamic behavior of all three designs was then simulated in a dedicated software (Proteus DS), considering time dependent forcing parameters (wind, currents and waves) to evaluate vertical load, components position, tilt and tension, required safe anchor mass, and overall mooring configuration, according to different set scenarios, i.e. "normal", "storm", and "extreme" (Figure 1). Moorings were required not only to endure "extreme" conditions without failure, but also maintain operational capabilities (to a reasonable extent) under more energetic events.

[Figure]

| daN | A (Buoy) | | B (Down-weight) | | C (Float I) | | D (Release) | |
|---|---|---|---|---|---|---|---|---|
| | Max | Avg | Max | Avg | Max | Avg | Max | Avg |
| "Normal" | 70 | 56 | 33 | 28 | 12 | 9 | 18 | 14 |
| "Storm" | 126 | 76 | 100 | 51 | 84 | 36 | 88 | 41 |
| "Extreme" | 335 | 120 | 317 | 100 | 302 | 86 | 306 | 91 |

*Figure 1. Sample of obtained results from Proteus DS according to defined scenarios for the vertical profiler mooring*

On the one hand, simulated results pointed to neglectable instrumentation tilt of the subsurface moorings under a set maximum 0.6 m/s current. On the other hand, the surface mooring was expected to be unable to maintain operational conditions in all simulated scenarios, where
* * *
[1] 1 Coppola, L., Ntoumas,M., Bozzano, R., Bensi,M., Hartman, S. E., CharcosLlorens, M. I., et al. (2016). Handbook of Best Practices for Open Ocean Fixed Observatories. European Commission, FixO3 Project, 127. (European Commission, FixO3 project, FP7 Programme 2007-2013 under grant agreement n° 312463). Available online at: http://hdl.handle.net/11\penalty-\@M329/302

vertical travel wire inclinations above 20º was expected to hinder vertical motion. Simulated inclinations were on average 5.7º, 15.2º, and 29.5º, under "normal", "storm" and "extreme" scenarios, respectively, as such considered to be satisfactory.

Reviewer: Side comment - lettering/labels in Figures could not be read.

Thank you for addressing formatting and style issues. Figures will be modified and refined in the revised manuscript.

Reviewer: Methods - how often are the moorings recovered and reset? R: Typically, 4 months each deployment. Are the new moorings installed to overlap the old moorings to allow for no gaps in the record? R: No. A gap will always exist between deployments. What conditions are the moorings/instruments in when recovered - biofouled, tangled with fishing line?

Recovered equipment experienced, as expected, biofouling, especially the salinity sensor of the vertical profile, whose data was discarded from this data paper. While in the subsurface platforms, placed broadly bellow 150 m depth, biofouling was not significant, the vertical profiler, was seriously affected. Although operating depths, between approximately 150m and 1m in constant motion (~5 or 6 full profiles per hour), should deter biofouling growth, after two months algae growth was significant, leading to sensor operation hinderance. Based on these findings, a new strategy must be developed for future deployments whether it is to clean the sensors regularly during the deployment period or use sensors less sensitive to biofouling (sensors with wipers for instance).

Deployment site selection carefully considered fishing activity in the surrounding area, avoiding well known heavy equipment preferred routes (e.g. trawling, longlines). However, coastal fisheries are an unpredictable challenge. To minimize this risk, engagement actions were undertaken with local communities, as well as acquiring a "no go" area through a legal concession, publicized in local navigation charts through official channels. This approach proved to be successful as there was no visible and/or reported incident, besides the single inquiry from a local fisherman about the surface buoy wondering if was lost

Currently there are no resources to exchange the set of equipment with secondary equipment during recovery. The authors are aware that it would be best practice to substitute the equipment during the period of recovery to guarantee continuous observation and if resources permit, the authors will proceed with this approach and revise the deployment plan accordingly. Recently IPMA (Portuguese Institute for the Sea and Atmosphere), the leading institution of the EMSO.PT consortium, acquired a similar wave powered vertical profiler, same brand (Wirewalker) and same payload. It is envisaged that such equipment will alternate with the CCMAR vertical profiler, to guarantee the permanent operation of the IbMa-CSV observatory, and continuous timeseries.

Reviewer: Is part of the quality control a comparison of successive deployments - in other words, making the sequential, successive records of similar quality and able to be merged? R: This was the first 4-month deployment. Yes, for future deployments. The authors attempt to include a quality control measure in which deployments shall be compared among one another as soon

another series of data is available. Therefore, exclusively standard quality control methods were applied to the dataset. Are shipboard observations made (like water sampling for in the lab analyses) made to check the instruments? R: The instruments came calibrated, directly from the manufacturer. Anyway, instruments were checked in lab with prepared solutions. Are recovered instruments post-calibrated before being cleaned and refurbished? R: Yes, they were already checked for calibration and ready for new deployment. Maybe the paper is presenting just one deployment - it is not clear. R: Yes, just the first deployment. The last sentence of $2^{nd}$ paragraph of Section 2 clarifies it. The abstract says this is an observatory so the reader assumed this is an ongoing long-term deployment with multiple cruises/mooring deployments. R: Yes, it is our goal.

Reviewer: Processing notes - is there a netcdf header for the files with processing information?For example, was a magnetic deviation correction applied to the velocity data and what was it?  Besides serial numbers, do you track and record software and firmware numbers and revisions for each instrument?  There looked to be no tracking of calibration information (history, date, coefficients....); where is that available?

The authors did not process the data to guarantee the provision of raw datasets for end-users. The quality control aims  to flag the data for the user to process the data upon their needs and requirements. Therefore, the authors refrained from any data alteration. The netcdf files provided comprehensive information about the data following the metadata standards from the EMSO-Eric observatory. No magnetic deviation correction was done for this data as the deployment site was only 90 km away from the calibration site. Calibration was done according to the manufacturer manual.

Reviewer: Figs 3 and 4 - hard to read, small labels.

Thank you for pointing out formatting and style issues. Figures will be modified and refined in the revised manuscript

Reviewer: Fig 6 - got very little out of this figure, small labels, faint contour lines, not the best color palette choices.

Thank you for pointing out formatting and style issues. Figures will be modified and refined in the revised manuscript

Reviewer: Section 5 Data Records - this is a place where knowing the goals of the observing effort would set the context. For example, is Fig 3 showing success or failure?  If the goal was full water column, result is a failure.  Be great to have mixed layer depth drawn on top of the velocity contours.  Figure 4 - does this address a goal, show success?

Due to the nature of this manuscript, it does not aim to achieve any specific research goals. The purpose of this manuscript is to openly share a high resolution (spatial and temporal) dataset following FAIR standards to facilitate data use for end-users. The observatory aims at collecting long-term and continuous datasets in the scope of the UN Ocean Decade Challenge 7 "Expand

the Global Ocean Observing System, which the authors will clarify in a goal section in the revised manuscript.

Reviewer: Table 3.Started to read this and realized there is not much information about the time bases of the files?  There is some instrument sampling info on page 6.  Were instrument clock time bases checked?  Were the raw sampling rates maintained in future files or were files taken to some common time base, like 1 hour?  What files and time base were used to prepare Table 3?  By showing four months, you open the question of whether or not there were statistically significant differences in these numbers month to month, but probably cannot address that due to small number statistics.

The information on page 6 explains the sampling rate for each sensor. This information could be repeated in the result section and in the figure labeling to facilitate the reading of this manuscript. The authors consider repeating this information in the result section. No parameter was rebased to a new time stamp and maintained in its original form to preserve raw data for the end user so they can adjust and process the data accordingly to their needs. The data represented in Table 3 was collected by the vertical profiler with a sampling rate of 2 Hz (0.5 seconds). The authors agree to add this information to the table description. Since the aim of this manuscript was not to answer a research question, the question whether this dataset is statistically significant or not is not essential. The value behind this data is the high-resolution information it holds in a highly active and interesting area as pointed out in the introduction. In a majority of studies lack of data is the limiting factor to establish reliable research, therefore this dataset contributes to an open-access and comprehensive FAIR data lake, which can be used by any user for their research.

Reviewer: Fig 8 is potentially very nice, but hard to read now with small font, poor color palette for BC frequency (N squared rather than N2).B great to see middle and lower panels with mixed layer depth overplotted.  How was spiciness computed?

Thank you for pointing out formatting and style issues. Figures will be modified and refined in the revised manuscript

Spiciness was computed with the Thermodynamic Equation of Seawater 2010 (TEOS-10) python package. This function calculates spiciness from Absolute Salinity and Conservative Temperature (pressure of 0 dbar) according to McDougall and Krzysik (2015), a routine "based on the computationally efficient expression for specific volume in terms of SA, CT and p" (Roquet et al., 2015).

Reviewer: The first part of the text in section 7 should have been in the introduction of a section 2, to explain the context and that a one deployment snapshot was being deployed. This section would also make more sense if the goals for observing had been defined early in the paper. R: As suggested by the reviewer, a Section 2 – Motivation and Objectives was introduced. There, the context of the observatory is described. Along with the 1. Introduction, where the oceanographic context is depicted and the importance of observing was shown, the relevance and uniqueness of the data is clarified in the last paragraphs of Section 2. Now, in Section 8 (former Section 7) the text in the initial part assumes that the data has already been presented, the reader already knows

them. For this reason, the authors think that now, with the present organization of the manuscript, it makes no sense to move this piece of text to Section 2. also reading the claim about observing mesoscale and submesoscale events makes the reader think there was some point in the velocity roses being shown that was not clear. R: As pointed out in the caption of Figure 4, the velocity roses is a demonstration of the overall data gathered by the ADCP. The analysis of mesoscale and submesoscale features must be done on appropriate subsets of the time series. But with basically a point observatory, how are you able to claim you sampled features defined by their spatial scales (submesoscale and mesoscale)? R: Our rational to write "spatial scales" was that as the observatory components are vertical profilers (Wirewalker and ADCP), the meaning of "spatial" was obviously the vertical spatial scale. In the entire manuscript, for clarity and when suitable, "spatial" was replaced by "vertical". Are you thinking of this in-situ data set in the context of satellite altimetry or some spatial sampling that would complement these point time series? R: The aim of this manuscript is to show, describe and characterise the first set of data from the IbMa-CSV observatory, that has been made available together with qc-flagging to guarantee its quality levels. It is not our intention in this data-paper to explore the scientific results that this data may reveal. To that end a separate science-paper is underway. With the publication of this in situ data, we intend to make available to the scientific community a high-resolution time series that has multiple applications, as we describe in the manuscript.

Reviewer: Conclusion: Statements like "crucial to understand the complexity of ocean dynamics..." are hard to accept as no ocean dynamics framework that motivated observing goals or no set of science driven questions that gave rise to specific observing requirements were presented. This would be a much better paper if that thought processes that showed the flow from science questions (for example, do eddies transport boluses of different water types or do eddies have associated enhanced vertical mixing) to observing requirements/goals was laid out up front. Then the paper could highlight the successes/failures of the observatory and move on to how the team will iterate to improve the observing effort. As a data presentation alone, it also needs some work and some better figures.

The IbMa-CSV observatory was not deployed to "test hypothesis". In that sense it was not installed to answer specific scientific questions. However, as described in the first two sections, it was installed in a location of intense and important mesoscale activity, consequence of multiple forcing factors acting. The mesoscale activity in the region could have effects throughout the North Atlantic. Thus, the location and the high-resolution nature of the data retrieved in the IbMa-CSV observatory were science driven. Due to its nature and uniqueness in a vast region from the Gulf of Cadiz to the Mid-Atlantic Ridge, if correctly used, the data meets the observing requirements to progress in the understanding of the complexity of the ocean dynamics in the region.

Moreover, the authors will consider to describe the observatory's significance of a high resolution data set as such presented in this manuscript in the context of the international strategy for the ocean by the European Union and UNESCO. Upon statements like "crucial to understand the complexity of ocean dynamics" will become acceptable since the scope of this observatory fits directly into the framework of the UNESCO Ocean Decade , in which a global observation network is a strong focus in the 10 Challenges. This observatory provides end-users with

valuable data information, leveraging a comprehensive global database to formulate and address research questions.

Reviewer: The report is an original and relevant attempt to establish high-resolution observations in the western northern margin of the Cape San Vicent. It reads well. Nevertheless the ms. could have a much broader scope/impact if it integrated more EMSO-ERIC observations. Consequently, the data is considered a limited subset (4 months). Is it possible to include other Regional partners and/or datasets?

The authors acknowledge the relevance attributed to the data presented in the MS. The present MS is intended to report only data sets from the Iberian Margin node of the EMSO-ERIC initiative. Data from the other regional partners will appear in the data page in the EMSO-ERIC site (https://emso.eu/data/)

Reviewer: Nevertheless, it represents a unique dataset with important uses for the scientific community. What about other users? Are these observations useful to major EU initiatives such as CMEMS? How? Can it complement satellite observations?

Our expectation in rapidly releasing the data through the SEANOE repository is that it can be used by any stakeholder, in specific research that needs the data, regardless of the use. Joint use with satellite observations should be explored, as this dataset has high temporal and vertical resolution, and satellite observations complement horizontal resolution, even if only at the surface. The suitability of this data for improving the parameterisation of numerical models is also highlighted in this MS.

Reviewer: The introduction reads like an introduction to the oceanography of the region, however, perhaps a more appropriate approach is to report on other historical observational efforts in the region, their limitations, and thus the importance of this new time series. Perhaps the Portuguese Hydrographic Institute; The European Marine Observation and Data Network (EMODnet); or the Environmental Observatory of the Strait of Gibraltar from CEIMAR could provide good complementary/permanent observations for an historical context. It is important to demonstrate the uniqueness of this dataset. Perhaps by drawing a timeline of all recorded observations in the region? More focus on the type, frequency, and limitations of historical observations and less on the oceanography.

The overview of the oceanography of the region is justified by the need to prove the choice of the region for the installation of the observatory and the usefulness of the time series of data in the context of the oceanographic processes taking place in the region. The authors agree that the approach could have been different, centered on the historical observational efforts in the region. Despite that, the authors decided to center the data paper on the relevance of the present

observations on the frame of the oceanographic context, that we consider a valuable approach too.

However, the authors agree with the reviewer that a better demonstration of the uniqueness of this dataset is needed. The authors took this opportunity to mention other observations in the region and their limitations, compared to the current set of observations, corresponding in this way to the suggestion of the reviewer. Reading Section 8 (former Section 7), the authors realize that the uniqueness of the observations was already mentioned, although without the appropriate emphasis. The section was slightly modified, and a new paragraph was added:

"The present data set, with such vertical and temporal resolution, is unique in the region. The nearest moorings, runed by the Instituto Hidrográfico (Portugal), are more than 120 km away, and only take measurements at the surface (https://www.hidrografico.pt/index/en). The velocity field is assessed only at the surface through HFRs that cover the region, runed by the Puertos del Estado (Spain) (https://www.puertos.es) and the Instituto Hidrográfico (Portugal). For the first time the subsurface is sampled in the region. The high-resolution sampling, covering the surface layer down to a depth of 150 meters, makes this dataset unique in a vast area of the ocean, disclosing the high oceanographic value of the data set."

Reviewer: Since the authors are not drawing any oceanographic conclusions from the dataset, the introduction does not match the main conclusions.

The introduction puts the scope of the observatory into context. The authors do not aim at drawing any conclusion due to the nature of this observatory and the resulting data paper, which goal it is to establish a continuous, longt erm and high-resolution ocean observatory and share the acquired data openly. The authors just made a demonstration of the value of the data for a suite of applications, based on the known oceanography of the region.

Reviewer: Data thresholds, data processing, and quality control were well addressed.

Reviewer: Figure legends are small and hard to read.

Thank you for pointing out formatting and style issues. Figures will be modified and refined in the revised manuscript